# Impact of Blueberry Consumption on the Human Fecal Bileacidome: A Pilot Study of Bile Acid Modulation by Freeze-Dried Blueberry

**DOI:** 10.3390/nu14183857

**Published:** 2022-09-17

**Authors:** William Gagnon, Véronique Garneau, Jocelyn Trottier, Mélanie Verreault, Charles Couillard, Denis Roy, André Marette, Jean-Philippe Drouin-Chartier, Marie-Claude Vohl, Olivier Barbier

**Affiliations:** 1Laboratory of Molecular Pharmacology, Endocrinology and Nephrology Axis, CHU de Québec-Université Laval Research Center, Quebec City, QC G1V 4G2, Canada; 2Faculty of Pharmacy, Université Laval, Quebec City, QC G1V 0A6, Canada; 3Institute of Nutrition and Functional Foods (INAF), Université Laval, Quebec City, QC G1V 0A6, Canada; 4Centre Nutrition, Santé et Société (NUTRISS), Université Laval, Quebec City, QC G1V 0A6, Canada; 5Department of Food Science, Faculty of Agriculture and Food Sciences, Université Laval, Quebec City, QC G1V 0A6, Canada; 6School of Nutrition, Faculty of Agriculture and Food Sciences, Université Laval, Quebec City, QC G1V 0A6, Canada; 7Department of Molecular Medicine, Faculty of Medicine, Université Laval, Quebec City, QC G1V 0A6, Canada; 8Quebec Heart and Lung Institute (IUCPQ) Research Center, Quebec City, QC G1V 4G5, Canada

**Keywords:** blueberries, bile acids, dietary supplements, polyphenols, LC-MS/MS profiling

## Abstract

Cholesterol-derived bile acids (BAs) affect numerous physiological functions such as glucose homeostasis, lipid metabolism and absorption, intestinal inflammation and immunity, as well as intestinal microbiota diversity. Diet influences the composition of the BA pool. In the present study, we analyzed the impact of a dietary supplementation with a freeze-dried blueberry powder (BBP) on the fecal BA pool composition. The diet of 11 men and 13 women at risk of metabolic syndrome was supplemented with 50 g/day of BBP for 8 weeks, and feces were harvested before (pre) and after (post) BBP consumption. BAs were profiled using liquid chromatography coupled with tandem mass spectrometry. No significant changes in total BAs were detected when comparing pre- vs. post-BBP consumption samples. However, post-BBP consumption samples exhibited significant accumulations of glycine-conjugated BAs (*p* = 0.04), glycochenodeoxycholic (*p* = 0.01), and glycoursodeoxycholic (*p* = 0.01) acids, as well as a significant reduction (*p* = 0.03) in the secondary BA levels compared with pre-BBP feces. In conclusion, the fecal bileacidome is significantly altered after the consumption of BBP for 8 weeks. While additional studies are needed to fully understand the underlying mechanisms and physiological implications of these changes, our data suggest that the consumption of blueberries can modulate toxic BA elimination.

## 1. Introduction

Cholic acid (CA) and chenodeoxycholic acid (CDCA) are the primary bile acids (BAs) formed from cholesterol in the liver, and stored in the gallbladder under the forms of conjugates to the amino acids glycine or taurine also known as bile salts [1]. Glyco- and tauro-conjugated BAs can then be secreted into the biliary tract to reach the intestinal lumen where they favor the absorption of fat-soluble nutrients [1]. Due to their amphipathic nature, BAs have the ability to form micelles, which promote the intestinal absorption of dietary lipids and other lipophilic compounds such as vitamins A, D, K, and E [2]. From the intestinal lumen, BAs are recycled through enterohepatic recirculation. The majority of BAs are actively reabsorbed in the ileum via captation by the apical sodium-dependent bile salt transporter from enterocytes [3]. Then, BAs can be secreted from the ileocytes to the portal circulation to return to the liver [3]. This mechanism, together with low passive absorption throughout the intestinal tract, is responsible for the effective recycling of 95% of the BAs secreted in the intestine [1]. The remaining 5% (roughly 500 mg per day) are excreted in feces [4]. BAs that are not reabsorbed undergo many transformations by the microbiota, most of which occur in the ileum and colon. Gut bacteria can deconjugate conjugated BAs via bile salt hydrolase enzymes to form unconjugated BAs, which can then be dehydroxylated into secondary lithocholic acid (LCA) and deoxycholic acid (DCA) via the 7α-dehydroxylase enzyme [5].

BAs and their derivatives play major roles in the regulation of cholesterol and act as endogenous ligands for several nuclear and membrane receptors (namely the farnesoid X receptor (FXR), pregnane X receptor (PXR), vitamin D receptor (VDR), constitutive androstane receptor (CAR), and Takeda G protein-coupled receptor 5 (TGR5)), to regulate functions such as glucose and lipid metabolism, inflammation, and immunity [1,4,6,7]. BAs display different capabilities for binding to and activating these receptors. For example, as an FXR agonist, the primary acid CDCA and its tauro-conjugates play a determinant role in the intracrine negative feedback loop in which BAs control their own hepatic synthesis [4]. By contrast, the secondary acid, glycoursodeoxycholic acid (GUDCA), can act as an FXR antagonist [8]. Intestinal secondary acids such as DCA and LCA also act as natural antibiotics and thus influence microbiota diversity [9]. In such a context, minimal changes in the BA pool composition can have a major impact on human health.

To ensure an optimal level for these biological functions, the formation and metabolism of BAs are highly and specifically regulated by endo- and exogenous factors such as negative feedback loops, feeding state, circadian rhythm, and enteric reabsorption [1]. In addition, the composition of the BA pool can be influenced by dietary factors, microbiota composition, metabolic disorders, and several pathologies of the gut–liver axis [10]. Such a complex regulation process results in different BA profiles, particularly following specific dietary intervention [11].

Dietary polyphenols are among the food components currently investigated in depth regarding human health are polyphenols [12,13]. Recent observations revealed that consumption of polyphenol-rich fruits can affect the bileacidome [13,14]. Blueberries are highly concentrated in anthocyanins, a type of polyphenol that give the characteristic colors of the berries [15] and are known to influence the microbiome [16,17]. Based on these observations, we hypothesized that the consumption of blueberry extracts may have significant impacts on the fecal bileacidome. To test this hypothesis, the fecal bile acid profile was measured before and after an 8-week consumption period of 50 g/day of freeze-dried blueberry powder (BBP) by in 24 human volunteers.

## 2. Methods

### 2.1. Ethics Statement

This study was approved by the Ethics Committees of Université Laval and the CHU de Québec Research Centre. The study is registered at https://clinicaltrials.gov/ (accessed on 8 May 2022) (NCT03266055). All subjects signed a written informed consent prior to their participation in the study.

### 2.2. Participants and Original Design

Samples analyzed in this study were originally collected from volunteers of a randomized, double-blind, placebo-controlled trial measuring the impact of BBP supplementation on immune related pathways (see reference [18]). Briefly, the original study was conducted between 2017 and 2019 at the Institute of Nutrition and Functional Foods (INAF) of Université Laval in the Quebec City area (Canada). Caucasian men and women (premenopausal), in good health, and aged 18–55 years were recruited. The original trial involved 49 participants randomized to either the placebo (*n* = 24) or BBP intervention group (*n* = 25).

From the 49 participants involved in the original trial [18], only those from the intervention group were selected for the present proof-of-concept study on bile acids. A sample from one participant in the BBP group could not be used as it was of insufficient quantity to be profiled for BAs. In summary, the final cohort involved in the present investigation comprised 24 Caucasian individuals (11 men and 13 premenopausal, nonpregnant and nonlactating women) of the BBP intervention group that were at risk of metabolic syndrome (Table 1).

As extensively described in Rousseau et al. [18], the inclusion criteria were a BMI of 25.0–40.0 kg/m^2^ or abdominal obesity (waist circumference ≥ 94 cm for men and ≥80 cm for women); triglycerides ≥1.35 mmol/L or fasting insulin concentration ≥ 42 pmol/L [19]. Participants were excluded if, prior to the study, they were diagnosed with diabetes, hypercholesterolemia, or hypertension, and if they were taking medications for these conditions. Participants were also excluded if they took antibiotics during or up to the months before the study [18].

### 2.3. Intervention

Participants were asked to consume 50 g of a freeze-dried BBP per day (2 doses of 25 g separated by at least 8 h) for 8 weeks. A total of 50 g BBP is roughly equivalent to 350 g of fresh blueberries [18]. The BBP was provided by the U.S. Highbush Blueberry Council (Folsom, CA). It consisted of a blend of milled freeze-dried highbush blueberries from two cultivars (*Vaccinium virgatum* (*ashei*) and *Vaccinium corymbosum*) in a 1:1 ratio [18]. Participants were asked to add the BBP to 300 mL of water or to mix it in food that would not compromise the phytochemicals [18]. They were instructed not to make major changes to their lifestyle or dietary pattern for the duration of the study. Restrictions were imposed on whole-berry consumption as well as on certain products with high contents of phytochemicals such as red wine, coca products, tea, and coffee [18].

At week 0 (pre-) and 8 (post-treatment), participants were asked to collect stool samples at home at a bowel movement as close as possible to the visit planned to the research center [18]. They had to keep the samples in their freezer (−20 °C) until they brought it to the research center. An icepack was provided to keep the samples frozen during transportation. Thereafter, stool samples were stored at −80 °C at the research center until BA profiling.

### 2.4. Measurement of Fecal Bile Acids

A total of 19 BA species were extracted, separated, and quantified using a previously reported liquid chromatography coupled with tandem mass spectrometry (LC-MS/MS) method [20,21]. Briefly, fecal samples were lyophilized under nitrogen and resolubilized in a water:methanol (50:50) solution containing 0.1% formic acid. Stainless-steel beads were added to the resulting suspension and blended at 4 °C to obtain a homogenate (Blender, Next Advance, NY, USA). Organic fraction was evaporated under nitrogen before resuspension in water containing 0.1% formic acid. Deuterated BAs were added as an internal standard. After centrifugation at 5000 g for 5 min, the supernatant was collected and the organic fraction was evaporated under nitrogen before resuspension in water containing 0.1% formic acid. The solution containing BAs then underwent a solid-phase extraction using Strata-X 60 mg columns (Phenomenex, Torrance, CA, USA). As extensively reported [20,21], the LC-MS/MS system consisted of a Nexera ultra-high-pressure liquid chromatography (UHPLC) instrument (Shimadzu Scientific Instruments, Columbia, MO, USA) coupled with an API6500 tandem mass spectrometer (Applied Biosystems, Concord, ON, Canada).

HPLC-grade solvents were purchased from VWR Canlab (Montréal, QC, Canada). Ammonium formate was bought from Laboratoire Mat (Quebec City, QC, Canada). Deutered BAs standards (d_4_-CA, d_4_-CDCA, d_4_-LCA d_4_-DCA, d_4_-GCA and d_4_-TCA) were purchased from C/D/N Isotopes (Montréal, QC, Canada) and Toronto Research Chemicals (Toronto, ON, Canada). The separation column for chromatographic separation used was a Poroshell 120 EC-C18 2.7 μm; 2.1 × 150 mm (Agilent, Santa Clara, CA, U.S.A.).

### 2.5. Bile Acid Analysis

As previously reported [21,22], total BA concentrations correspond to the sum of all BAs measured. Total sums of glyco- and tauro-conjugates were calculated by the summation of concentrations of conjugated BAs: CDCA, CA, DCA, LCA, UDCA, and hyodeoxycholic acid (HDCA). The sum of unconjugated BAs also included HCA concentration. The total of primary, secondary, and hydroxylated BA species was determined by adding all unconjugated and/or conjugated species of CDCA + CA, LCA + DCA, or HDCA + hyocholic acid (HCA), respectively. The total of CA, CDCA, LCA, DCA, HDCA, or HCA was obtained by summing the concentrations of all forms (unconjugated and conjugated) for each species.

### 2.6. Statistical Analysis

All data are presented as mean ± SEM. Bile acid levels showed neither normal nor lognormal distribution using normality and lognormality hypothesis tests (Anderson-Darling test, D’Agostino and Pearson test, Shapiro–Wilk test, and Kolmogorov–Smirnov test) and failed visual test for normality with a quantile–quantile plot. Thus, nonparametric tests were chosen for all the analyses. To assess differences in BA levels after the intervention compared with baseline, the Wilcoxon matched-pairs signed rank tests were also corrected for multiple comparisons using the Bonferroni–Dunn method. For differences between sex in BA profiles, the Mann–Whitney test corrected for multiple comparisons with the Bonferroni-Dunn method was used. Descriptive statistics (mean, standard error of the mean (SEM), coefficient of variation (CV), range, etc.) are used to present the data and to characterize interindividual variation. All statistical analyses were made using Prism 9 from GraphPad Software™ (San Diego, CA, USA). The significance threshold was set at *p* values < 0.05 (two sided).

## 3. Results

### 3.1. Fecal Bile Acid Profiles Sustain Large Inter-Individual Variability

As illustrated in Table 2, unconjugated BAs were the main component of the fecal pool, representing 95.4 ± 0.7% at the beginning of the study. Most of the unconjugated species corresponded to secondary acids, which represented 90.2 ± 2.0% of the total BA pool (Table 2). With respective levels of 2.91 ± 0.31 and 2.53 ± 0.23 nmol/mg of feces, the unconjugated forms of the secondary acids, DCA and LCA, were the most abundant species (Table 2). Glycine- and taurine-conjugated BAs together represented only 1.75 ± 0.52% of fecal BAs.

At week 0 (Appendix A), the initial profile of BAs showed a large interindividual variability in both male and female volunteers. The total BA concentration ranged from 0.47 to 12.29 nmol/mg of feces (CV: 62.1%) in men and from 3.71 to 10.20 nmol/mg of feces in women (CV: 34.6%) (Appendix A). The largest variations were observed with glycine-conjugated species in men (CV: 177.0%) and with taurine-conjugated species in women (CV: 182.3%) (Appendix A). An important variation was also observed among CA species for both sexes, with total CA values ranging from 0.01 to 0.69 nmol/mg of feces in men (CV: 160.0%) and from 0.01 to 1.64 nmol/mg of feces in women (CV: 175.5%). The less variable parameters were the 6a-hydroxylated BAs in men and the total LCA species in women, with values ranging from 0.0007 to 0.0199 nmol/mg of feces (CV: 56.6%) and from 1.67 to 4.24 nmol/mg of feces (CV: 32.9%), respectively.

### 3.2. No Sexual Dimorphism Was Detected in the Fecal Bileacidome

No significant changes were detected between men and women concerning the fecal BA profiles in samples harvested before (Appendix A) and after (Appendix A) the BBP consumption period.

### 3.3. Fecal Bileacidome Exhibit Significant Alteration after 8-Week Consumption of Freeze-Dried Blueberry Powder

As summarized in Table 1, the 8-week consumption period of BBP led to an altered fecal BA profile. While the changes in specific parameters (namely GCDCA, GUDCA, and the sum of glycine-conjugated and secondary acids) reached statistical significance (Figure 1), most of the parameters remained either unaffected or only tended to be modified in a nonsignificant manner (Table 2). For example, the reduction in total BA levels from 6.01 ± 0.57 to 4.84 ± 0.46 nmol/mg of feces failed to reach statistical significance (Table 2 and Figure 1A). The same also applied to unconjugated bile acids (Figure 1B). By contrast, the level of glycine-conjugated BAs was significantly, 1.5-times, increased in feces harvested after the diet compared with initial levels (Figure 1C). This increase reflected a significant accumulation of GCDCA (*p* = 0.010) and GUDCA (*p* = 0.011) (Figure 1D,E), while the 1.75-fold increase in GCA fecal levels remained nonsignificant (*p* = 0.115) (Table 2 and Figure 1F).

Beyond conjugated and unconjugated forms, BA species can also be divided into primary and secondary species. While the increase from 0.33 ± 0.11 to 0.49 ± 0.25 nmol/mg of feces in primary BAs remained statistically nonsignificant (Figure 1G), the reduction in the levels of secondary BAs (from 5.48 ± 0.52 to 4.15 ± 0.37 nmol/mg of feces) observed in post-BBP samples reached statistical significance (*p* = 0.030; Figure 1H). Interestingly, the fecal concentrations of secondary DCA and LCA acids also tended to be reduced (from 2.91 ± 0.32 to 2.13 ± 0.23 nmol/mg of feces for DCA and from 2.53 ± 0.23 to 1.98 ± 0.16 nmol/mg of feces for LCA) after BBP consumption (Figure 1I,J; *p* = 0.103 for both metabolites).

Overall, these observations indicated that BBP treatment significantly altered the fecal BA profile.

## 4. Discussion

In this pilot study, significant alterations of the human fecal bileacidome were detected after an 8-week consumption of BBP. The most significant changes corresponded to increased concentrations of glycine-conjugated BAs and a decrease concentration of secondary BAs.

Profiling the prediet samples revealed that secondary acids such as LCA and DCA dominated the fecal BA profile, which is in accordance with the findings of previous reports [23]. An important interindividual variability was observed both at baseline and at the end of the study for all BAs. This important interindividual variability in fecal BA profiles is also consistent with findings of previous studies in overweight patients at high risk of metabolic syndrome [24]. No significant sex-dependent differences were detected either in the pre- and post-BBP samples. While such differences were previously reported for the circulating BA profile [25], the present observations are consistent with the lack of studies reporting that sex affects the fecal excretion of BAs [26]. Parameters such as the changes in DCA and LCA levels tended to be reduced but failed to reach statistical significance, which was likely due to insufficient statistical power. It can be envisioned that a study with a larger number of volunteers might find significant differences in these parameters. Because BAs secreted from the liver substantially change with the intestinal microbiota [27], the above-mentioned changes in the fecal BA profile may reflect a modulation of the intestinal microbiota composition. Blueberries are particularly rich in anthocyanins, a subclass of flavonoid polyphenolic compounds with a characteristic three-ring structure [28,29]. A recent meta-analysis showed that anthocyanins alter the ratio of two major Bacteria phyla present in the human microbiota (i.e., Firmicutes and Bacteroidetes) [30]. Bacteria from these phyla are highly involved in the microbiota-dependent modifications of BAs in the intestine [31,32]. It can be hypothesized that the accumulation of glyco-BAs and the reduction in secondary acids reflect an altered relative abundance of bacteria belonging to Firmicutes and Bacterioidetes phyla. Beyond the modification of microbiota composition, polyphenols can also stimulate BA excretion through the formation of complexes with amidated BAs (i.e., glycine- and taurine-conjugated acids) [33]. Thus, additional investigations are warranted to not only further ascertain the role of polyphenols in the changes detected in post-BBP samples but also decipher the mechanisms governing such changes.

A second plausible explanation for these changes relates to the high levels of dietary fibers present in blueberries [34]. Previous results from the same study revealed that consumption of BBP significantly increased daily dietary fibers consumed by the participants [18]. Dietary fibers enhance fecal BA excretion (reviewed in [35]). While the underlying mechanisms beyond such enhancement remain to be fully elucidated, a recent review article by Naumann et al. [34] proposed two potential mechanisms: (1) fibers may cause an increased viscosity of the intestinal content that interferes with the micellar properties of BAs; and (2) fibers may form hydrophobic interactions that complex BAs to plant compounds. Both mechanisms may increase the fecal BA content via the hydrophobic properties of specific BA species [34]. Glycine-conjugated BAs, such as GUDCA and GCDCA, which were significantly altered after the post-BBP diet, are among the most hydrophobic BAs [36].

While we cannot exclude the possibility that the reported changes in the BA profile could also be due to other secondary effects of the intervention, we are confident that alterations in the fecal bileacidome were actually reflecting the impact of the BBP consumption, because each individual was compared with its own baseline. In addition, all participants were carefully instructed not to change their usual lifestyle habits.

Beyond their role in cholesterol elimination and intestinal absorption, BAs also play numerous roles as endocrine regulators, but their amphipathic nature can have toxic impacts when they accumulate at high concentrations [4]. Thus, we speculate that the alterations in the bileacidome detected in this pilot study may translate into important effects for human health. For example, GCDCA promoted liver fibrosis in experimental models of hepatocellular cholestasis [37], and a blueberry-enriched diet could be viewed as a mean to reduce liver fibrosis in cholestatic patients. Conversely, GUDCA was proposed as being part of the mechanism through which metformin improves obesity-induced glucose intolerance and insulin resistance [8]. The post-BBP diet reduction of this acid could be considered as a potential drug–diet interference for metformin-treated patients. In the same vein, post-BBP samples tended to present lower levels of the secondary DCA and LCA. While LCA is a pro-inflammatory molecule in the intestine, some of its derivatives produced by bacteria exert important immunomodulating effects in the intestinal epithelium and act as promoters of the intestinal barrier integrity via interactions with TGR5 [38]. A similar ambivalence also applies to DCA, the levels of which are associated to colon cancer, inflammation, and cell proliferation, but which is a potent antibiotic protecting against the growth of opportunistic bacteria known to cause health problems such as *Clostridioides difficile* [39]. In summary, how the changes detected in the BA profiles after BBP consumption will have beneficial or deleterious consequences for human health is likely to be the subject of important interindividual variability depending on the subject’s health status.

## 5. Conclusions

In conclusion, the results of this pilot study indicated that the fecal bileacidome of participants at high risk of metabolic syndrome was significantly influenced following an 8-week intake of freeze-dried BBP. Both the mechanisms leading to such modifications and their consequences for human health require additional investigation with larger populations and specific study designs to be fully understood. For example, future investigations will be designed to measure the longitudinal dynamics in bileacidome modifications and to investigate the contribution of gut microbiome to changes in the bileacidome. Nevertheless, our work suggests that the consumption of blueberries can be considered as a potential means to increase toxic BAs elimination.

## Figures and Tables

**Figure 1 nutrients-14-03857-f001:**
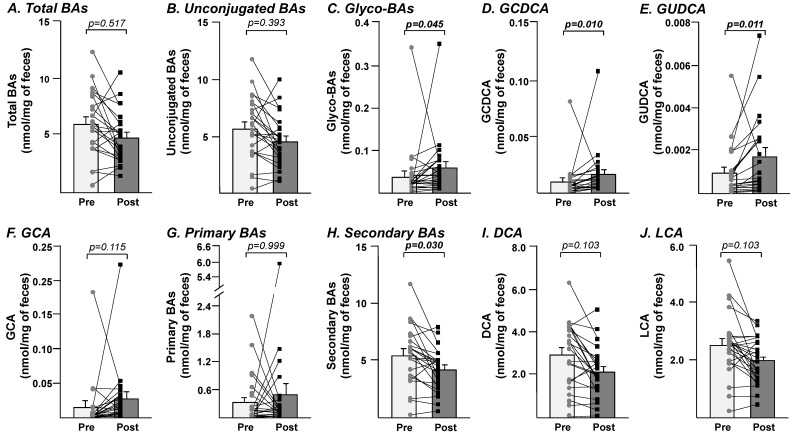
Fecal samples from the post-freeze-dried blueberry powder consumption period displayed significant accumulations of glycine-conjugated bile acids, GCDCA and GUDCA, while secondary bile acids levels were reduced. Bile acids from pre- and post-study samples were profiled using LC-MS/MS measurements, and bile acids composition analyses were performed as detailed in the Method section. Bars represent the mean ± SEM of the pre- (light grey) and post-BBP (dark grey) concentrations of total bile acids (**A**), unconjugated bile acids (**B**), glycine-conjugated bile acids (**C**), GCDCA (**D**), GUDCA (**E**), GCA (**F**), primary (**G**) and secondary (**H**) bile acids, as well as DCA (**I**) and LCA (**J**). Individual values are plotted and linked for baseline (circle) and after 8 weeks of treatment (square). Lines represent the specific trend for each individual. Statistical significance of the differences was assessed using Wilcoxon matched-pairs signed rank test and then adjusted for multiple comparisons using the Bonferroni-Dunn method and are expressed as adjusted *p* values. BAs; bile acids; glyco-BAs: glycine-conjugated bile acids; GCDCA: glycochenodeoxycholic acid; GUDCA: glycoursodeoxycholic acid; GCA: glycocholic acid; DCA: deoxycholic acid; LCA: lithocholic acid.

**Table 1 nutrients-14-03857-t001:** General baseline characteristics of participants from whom fecal samples were used in the present study. Adapted from Rousseau et al. [18].

Parameter		Mean	[Range]
Weight (kg)		89.5	[62.5–126.5]
Height (m)		1.71	[1.53–1.88]
BMI (kg/m^2^)		30.8	[23.4–47.1]
Waist circumference (cm)		101.5	[81.0–131.3]
Hip circumference (cm)		110.2	[92.2–141.8]
SBP (mmHg)		115.2	[99.0–131.7]
DBP (mmHg)		72.5	[57.0–82.3]
Apo B (g/L)		0.91	[0.60–1.41]
Total-C (mmol/L)		4.56	[2.99–6.41]
TG (mmol/L)		1.54	[0.53–3.98]
HDL-C (mmol/L)		1.17	[0.74–1.99]
LDL-C (mmol/L)		2.68	[1.58–4.05]
Total-C/HDL-C		4.06	[2.42–6.97]
HbA1c		0.051	[0.045–0.056]
Age (year)		35.0	[23.0–53.0]
**Sex**	**[*n*]**		
Men	11		
Women	13		
**Annual household income (CAD)**	**[*n*]**		
0–39,999	7		
40,000–79,000	8		
80,000–99,000	4		
≥100,000	4		
Nondisclosed	1		
**Highest education level completed**	**[*n*]**		
High school	2		
College	6		
University	16		

BMI, body mass index; Waist circ, waist circumference; Hip circ, hip circumference; SBP, systolic blood pressure; DBP, diastolic blood pressure; ApoB, Apolipoprotein B; Total-C, total cholesterol; TG, triglycerides; HDL-C, HDL cholesterol; LDL-C, LDL cholesterol; HbA1c, glycated hemoglobin; *n*, number of participants.

**Table 2 nutrients-14-03857-t002:** Bile acid concentrations before and after an 8-week period of consumption of freeze-dried blueberry powder.

	Before (Baseline)		After (Week 8)		Mean Diff.	Adjusted *p* Value
Bile Acids	Mean	SEM	Mean	SEM		
CA	0.1887	±0.0734	0.2732	±0.1672	0.0844	>0.999
CDCA	0.0967	±0.0278	0.1296	±0.0673	0.0328	>0.999
DCA	2.9097	±0.3159	2.1280	±0.2336	−0.7817	0.103
LCA	2.5275	±0.2297	1.9844	±0.1574	−0.5431	0.103
HDCA	0.0087	±0.0011	0.0064	±0.0008	−0.0023	0.375
HCA	0.0023	±0.0007	0.0019	±0.0006	−0.0005	>0.999
UDCA	0.0323	±0.0093	0.0518	±0.0192	0.0195	>0.999
GCA	0.0159	±0.0076	0.0279	±0.0090	0.0120	0.115
**GCDCA**	0.0102	±0.0032	0.0170	±0.0043	0.0068	**0.010**
GDCA	0.0132	±0.0031	0.0145	±0.0018	0.0013	>0.999
GLCA	0.0003	±0.0001	0.0003	±0.0000	−0.0001	>0.999
**GUDCA**	0.0010	±0.0002	0.0017	±0.0004	0.0008	**0.011**
TCA	0.0153	±0.0081	0.0212	±0.0098	0.0059	>0.999
TCDCA	0.0073	±0.0030	0.0188	±0.0095	0.0115	>0.999
TDCA	0.0306	±0.0117	0.0203	±0.0065	−0.0103	>0.999
TLCA	0.0015	±0.0007	0.0011	±0.0004	−0.0004	>0.999
TUDCA	0.0007	±0.0004	0.0014	±0.0008	0.0007	>0.999
TOTAL BA	6.0068	±0.5721	4.8358	±0.4598	−1.1710	0.517
Unconjugated	5.7660	±0.5556	4.5752	±0.4564	−1.1908	0.393
Taurine-conjugated	0.0554	±0.0228	0.0628	±0.0231	0.0074	>0.999
**Glycine-conjugated**	0.0406	±0.0139	0.0614	±0.0141	0.0208	**0.045**
Primary	0.3343	±0.1075	0.4877	±0.2482	0.1534	>0.999
**Secondary**	5.4828	±0.5251	4.1486	±0.3707	−1.3343	**0.030**
6α-hydroxylated	0.0110	±0.0015	0.0083	±0.0008	−0.0027	0.940
Total CA	0.2200	±0.0820	0.3223	±0.1735	0.1023	>0.999
Total CDCA	0.1144	±0.0299	0.1659	±0.0760	0.0515	>0.999
Total DCA	2.9536	±0.3213	2.1632	±0.2368	−0.7905	0.115
Total LCA	2.5294	±0.2299	1.9859	±0.1574	−0.5435	0.103
Total HDCA	0.0087	±0.0011	0.0065	±0.0008	−0.0023	0.414
Total HCA	0.0023	±0.0007	0.0019	±0.0006	−0.0004	>0.999

Values are presented as mean concentration (nmol/mg of feces) of the 24 pre- and postdiet samples ± SEM (standard error of the mean). Mean Diff., the difference between pre- vs. post-treatment were calculated for each of the participants (11 men and 13 women), and values represent the mean ± SEM. *p*-values were calculated using Wilcoxon matched-pairs signed rank test adjusted for multiple comparisons using the Bonferroni–Dunn method. Bold: *p* < 0.05. Bile acid composition analyses were performed as detailed in the materials and method section. CA: cholic acid; CDCA: chenodeoxycholic acid; LCA: lithocholic acid; DCA: deoxycholic acid; HDCA: hyodeoxycholic acid; HCA: hyocholic acid; UDCA: Ursodeoxycholic acid. G: glyco; T: tauro.

## Data Availability

All data could be obtained upon request to the authors.

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
