# Peer review of "Impact of Blueberry Consumption on the Human Fecal Bileacidome: A Pilot Study of Bile Acid Modulation by Freeze-Dried Blueberry"

_nutrients, 2022, doi:10.3390/nu14183857_

Round 1

Reviewer 1 Report

This study provides basic information on the effect of diet on BAs. The sample size is ok, and the presentation of the results is clear. Here are my concerns. 

(1)   The authors display the changes in BAs between the pre and post-stages. However, the essential information is neglected. The longitudinal dynamics (from weeks 0, 1,2,3......8) will be more informative: not just Pre VS. Post.

(2)   I am curious about the changes in the gut microbiomes. What’s the contribution of the gut microbiomes? Thus, I think, at least, the authors could add the changes in the gut microbiome and make some correlation analysis between the specific microbial group and metabolites, e.g., BAs.

(3)   I would appreciate it if the author could make the additional analysis on the effects of ages or the interaction effects of ages, sex, and other factors. Also, the tables or figures of these tests could be displayed in the maintext.

(4)   In figure 1, there are some outlier points. How to deal with these

Author Response

See the attache document

Reviewer 2 Report

This is an interesting study that explores a topic that has received little attention. Although it is a pilot study it would be good to state the primary hypothesis to be tested clearly, to avoid giving the impression that this is a fishing trip.

Round 2

Reviewer 1 Report

I have on other comments.